# CircuitNet: A Generic Neural Network to Realize Universal Circuit Motif Modeling

## Abstract

The successes of artificial neural networks (ANNs) are largely attributed to mimicking the human brain structures. Recent advances in neuroscience revealed that neurons interact with each other through various kinds of connectivity patterns to process information, in which the common connectivity patterns are also called circuit motifs. However, many existing ANNs can only model one or two circuit motifs in their architectures, so that their performance may drastically vary among different types of machine learning tasks. In this paper, we propose a new type of neural network inspired by the architectures of neuronal circuits, namely Circuit Neural Network (CircuitNet). In CircuitNet, a group of densely connected neurons, namely circuit motif unit (CMU), form the basic unit of the network, which is capable of modeling universal circuit motifs by adjusting the weights within the CMUs. Compared with traditional feed-forward networks, CircuitNet has the ability to model more types of neuron connections such as feed-back and lateral motifs. Inspired by the locally dense and globally sparse structure of the human brain, several iterations of signal transmission among different CMUs are achieved by sparse connections through the input ports and output ports of different CMUs. Experiments have demonstrated that CircuitNet can outperform popular neural network architectures in function approximation, reinforcement learning, image classification, and time series forecasting tasks.

## 1 Introduction

In the past decades, artificial neural networks (ANN) (McCulloch & Pitts, 1943) have been widely used as function estimators to solve regression and classification problems, which massively push deep learning forward in vastly different fields such as computer vision (He et al., 2016), natural language processing (Vaswani et al., 2017), deep reinforcement learning (Mnih et al., 2015), etc. The successes of ANN are largely attributed to mimicking the simplified human brain structures. For example, the original multi-layer perceptrons (MLP) are collections of neurons organized as layers (Minsky & Papert, 1969; Rosenblatt, 1958), and signals are controlled and transmitted between layers via linear transformations and non-linear activation functions, just like the synapses in a biological brain. Recently, the network architectures become rather complex, but their basic units such as convolutional layers and recurrent layers are still different abstractions of human nervous systems (Fukushima & Miyake, 1982; Hubel & Wiesel, 1968; Lindsay, 2021; Kietzmann et al., 2019; van Bergen & Kriegeskorte, 2020).

In recent years, understanding on both deep learning and neuroscience has made great advance, and it is time to rethink how artificial neural network designs can be further inspired by neuroscience. Following previous ANNs, we simplify the signal from a neuron as a real number, and focus more on how to model the signal transmission and connectivity patterns among neurons. Recent findings in neuroscience (Luo, 2021; Peters, 1991; Standring, 2021; Swanson, 2012) emphasized the role of specific patterns of synaptic connectivity in neuron communication across different brain regions. These patterns are analogous to the connection of neurons in ANNs, and in the rest of this section we will introduce how these findings can inspire the ANN design.

One line of advances in neuroscience revealed that neurons interact with each other through various kinds of connectivity patterns, namely the circuit motifs (Luo, 2021). There are four types of most common circuit motifs, including feed-forward excitation and inhibition, feed-back inhibition,

lateral inhibition, and mutual inhibition, as shown in Figure 1. However, neurons in most existing ANNs (including those with residual connections (He et al., 2016)) that incorporate merely feed-forward networks can only function as the feed-forward excitation and inhibition patterns. Even if some recurrent structures, e.g., RNNs (Hochreiter & Schmidhuber, 1997; Cho et al., 2014), are able to model cyclic patterns, they may fail with complex interactions among upstream neurons before information comes in.

Another typical property of the biological nervous system is the local density and global sparsity (Swanson, 2012). Although a single neuron can have thousands of synapses, most of them are located within a small region and form a functional group for specific tasks (Hawkins & Ahmad, 2016). Only a few of them travel to other functional groups as bridges between regions (Peters, 1991; Standring, 2021; Swanson, 2012). In conclusion, the biological nervous system has a locally dense and globally sparse connection structure. This is typically not true in ANNs where if we consider the vectors in a network as functional groups with some specific meanings, connections within a group are mostly missing but those between groups are of high density. For instance, the attentions between vectors in the original type of transformer form a complete graph and are of the highest density (Vaswani et al., 2017). On the other hand, many parameters within a neural network have been proved to be redundant (Glorot et al., 2011; Schwarz et al., 2021; Wen et al., 2016; Yoon & Hwang, 2017) and the global sparsity of the biological neurons can provide insight for us to reduce the complexity of the neural network (Mocanu et al., 2018; Pessoa, 2014; Strogatz, 2001).

As a result, the design of our proposed ANN follows two principles that previous work does not fully consider: 1) the low-level interaction pattern among neurons should contain different types of circuit motifs (universal circuit motifs), and 2) neuron connections should follow a locally-dense and globally-sparse pattern to form the high-level structure. In this paper, we propose a new type of architecture named Circuit Neural Network (CircuitNet) for neural network design. CircuitNet uses densely connected neurons as basic functional structure called Circuit Motif Unit (CMU) to support universal circuit motifs. The communication among different CMUs is achieved by the sparse connections between subsets of special neurons in CMUs, namely input/output ports. Experiments on both synthetic and real-world datasets prove that, with comparable or even fewer parameters, CircuitNet can outperform popular neural network architectures in various types of tasks, demonstrating its effectiveness and generalizability in machine learning. Notably, CircuitNet outperforms multi-layer perceptron in function approximation, image classification and reinforcement learning tasks, outperforms MLP-based, CNN-based and Transformer-based neural networks in image classification tasks on CIFAR-10, CIFAR-100 and ImageNet datasets, and outperforms CNN-based, RNN-based and Transformer-based neural networks in time series forecasting tasks.

## 2 CIRCUIT NEURAL NETWORK

In this section, we introduce the detailed implementation of the proposed Circuit Neural Network, including how to model the signal transmission between neurons in a Circuit Motif Unit (CMU) to support universal circuit motifs and the overall locally dense globally sparse connection structure.

### 2.1 NEURON SIGNAL TRANSMISSION FOR UNIVERSAL CIRCUIT MOTIFS

In this paper, the signal from a neuron is simplified as a real number. Given two or more connected neurons, their interaction is achieved by signal transmission, which is modeled as a transformation function $g(\cdot; \cdot)$. As shown in Figure 1, we aim at designing a transformation function that can model four types of neuron connection patterns (i.e., circuit motifs), including feed-forward, mutual, feed-back and lateral motifs, corresponding to the four patterns in recent biological findings (Luo, 2021). In feed-forward and mutual motifs, the input signal depends merely on the state of the previous neuron, while in the other two motifs, an input signal may be affected by the state of another neuron or the target neuron itself. In the following paragraphs, we present the design of our proposed transformation function and elaborate on why this design can model different circuit motifs.

**Linear Transformation** One of the simplest transformation used to pass signals from neuron $x_j$ to $x_i$ can be written as:

$$g(x_i; x_j) = W_{ij}x_j. \tag{1}$$

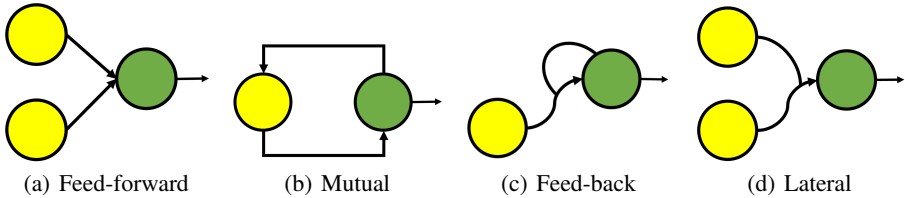

(a) Feed-forward       (b) Mutual       (c) Feed-back       (d) Lateral

Figure 1: Different motifs among neurons (Luo, 2021). A neuron in green is the target neuron in a motif and is updated in every iteration based on the signals from other neurons.

$W_{ij}$ is a learnable parameter indicating the intensity of influence from $x_j$ to $x_i$. Most of the previous ANNs also follow the idea of linear transformation because it is straightforward yet effective at modeling feed-forward and mutual motifs where input signals to a neuron are considered independently and added up simply.

**Neuron-wise Attention** Linear transformation modeling the feed-forward and mutual motifs assumes that the influence of an input signal is independent with other signals. However, this is sometimes over-simplified since there are also feed-back and lateral connections in human brains where the intensity of the input signal may be influenced by signals from other neurons and the target neuron itself. To equip CircuitNet with the ability to model these more complicated patterns, we need to find some transformations which take the interactions between input signals into consideration.

One concept we can borrow from traditional ANNs to model the interactions is the self-attention mechanism (Vaswani et al., 2017), but in CircuitNet, it is calculated between neurons (scalars) rather than groups (vectors). Therefore, we name it neuron-wise attention and formulate it as:

$$g(x_i; x_j) = \alpha_{ij}\mathbf{V_j}\mathbf{X}, \tag{2}$$

$$\alpha_{ij} = \operatorname*{softmax}_{j}((\mathbf{Q_i}\mathbf{X})(\mathbf{K_j}\mathbf{X})) = \frac{\exp((\mathbf{Q_i}\mathbf{X})(\mathbf{K_j}\mathbf{X}))}{\sum_k \exp((\mathbf{Q_i}\mathbf{X})(\mathbf{K_k}\mathbf{X}))}. \tag{3}$$

$\mathbf{X}$ contains the states of all the $M$ neurons in a group of connected neurons and $\mathbf{V_i}, \mathbf{K_i}, \mathbf{Q_i} \in \mathbb{R}^{1 \times M}$ are learnable linear projection weights for calculating value, key and query (which are 3 scalars) for the $i$-th neuron.

This type of transformation can be used to model the feed-back motif well because the influence of the input signal is adjusted by the target neuron state.

**Product between Neurons** Another way to model the interactions between input signals is to directly calculate the product of every pair of neurons, i.e.,

$$g(x_i; x_j, x_k) = \frac{1}{2}W_{ijk}x_j x_k, \tag{4}$$

where $W_{ijk}$ is a learnable parameter to model the intensity of the combined influence from $x_j, x_k$ to $x_i$. It is divided by 2 to eliminate the influence to the gradient caused by the quadratic term. Note that $j = i$ or $k = i$ is allowed in this formula so that the lateral motif (input signal is affected by another neuron) and the feed-back motif (input signal is affected by the target neuron) can be modeled at the same time.

In order to model the universal circuit motifs, different transformations will be used together to get a more comprehensive depiction of the circuit motifs. For example, we can use the product between neurons and the linear transformation together and write the transformation function as below:

$$g(x_i; x_j, x_k) = \frac{1}{2}W_{ijk}x_j x_k + V_{ij}x_j + V_{ik}x_k. \tag{5}$$

In this way, all four patterns illustrated in Figure 1 can be modeled properly.

### 2.2 Locally Dense and Globally Sparse Neuron Connections

In the previous section, we have introduced how CircuitNet models the signal transmission among the connected neurons. Now we introduce how neurons are connected in the proposed CircuitNet.

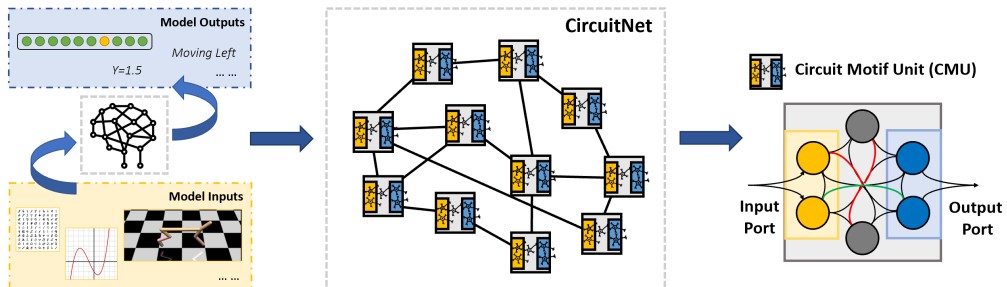

Figure 2: CircuitNet architecture. As a generic neural network, CircuitNet can be used in different tasks, take different forms of inputs and give corresponding outputs (left). It consists of several Circuit Motif Units (CMUs) connected with each other through their ports and maintains a global sparsity (middle). Within a single CMU, neurons are interconnected densely in a complicated manner in order to model a universal circuit motif, e.g., the green line connecting two neurons can be a linear transformation and the red lines can be a transformation involving 3 neurons (right).

**Circuit Motif Unit (CMU)**    Following the locally-dense and globally-sparse principle, CircuitNet contains clusters of neurons forming several densely connected neural circuit groups as its basic units, namely Circuit Motif Units (CMU), as $\mathcal{X} = \{\mathbf{X_1}, \mathbf{X_2}, \ldots, \mathbf{X_N}\}$ with $N$ units and $\mathbf{X_i} = (x_{i1}, x_{i2}, \ldots, x_{iM})^\top$ with $M$ neurons per unit. By applying the transformation function introduced in the previous section and dense connection, CMU is able to model all types of circuit motifs, i.e., universal circuit motifs. These CMUs are then sparsely connected by their input/output ports. The overall architecture of CircuitNet and the structure of CMU are shown in Figure 2.

As a result, after initialized by the task-specific input, different CMUs cooperate with each other by a certain number of iterations of intra-unit and inter-unit signal transmission. Specifically, when we look at a single unit, its state is updated based on two sources of signal input: the internal update among neurons within the CMU with the aforementioned transformations, and the received signals from other CMUs with edges to it, as follows:

$$\mathbf{X_i}^{(k)} = f(g_i^{(k)}(\mathbf{X_i}^{(k-1)}; \mathbf{X_i}^{(k-1)}) + \textstyle\sum_j^{e_{ji}\in\mathcal{E}} h_{ij}^{(k)}(\mathbf{X_j}^{(k-1)}) + \mathbf{b_i}^{(k)}), \tag{6}$$

where $\mathcal{E}$ denotes the set of connected CMU pairs. An edge $e_{ji} \in \mathcal{E}$ means the $i$-th CMU will receive signals from the $j$-th CMU at every iteration. $\mathbf{X_i}^{(k)}$ denotes the state of the $i$-th CMU at iteration $k$. $g_i^{(k)}(\cdot;\cdot)$ is the transformation function as described in Section 2.1 to model the universal circuit motif, $h_{ij}^{(k)}(\cdot)$ is the transformation function to process the signals from other CMUs, and $f(\cdot)$ is the normalization and activation function. $\mathbf{b_i}$ is the bias term. To simplify the notation without loss of generality, $i, k$ may be omitted elsewhere in the formulas when we are unconcerned about a specific CMU or iteration.

**Inter-unit Signal Transmission**    To enable inter-unit interactions by signal transmission, a specific sub-set of neurons from the CMU are responsible for sending out signals based on their own states, and another set for receiving signals from others. We call these neurons input and output ports, respectively. In this way, the update of neurons in a CMU through ports can be written as:

$$h_{ij}(\mathbf{X_j}) = \mathbf{P_{in,i}} \circ (\mathbf{W_{ji}}(\mathbf{P_{out,j}} \circ \mathbf{X_j})). \tag{7}$$

Here, $\mathbf{W_{ji}}$ is a learnable linear transformation from the $j$-th CMU to the $i$-th. $\mathbf{P_{in}}, \mathbf{P_{out}}$ are binary vectors with $p$ non-zero elements, identifying which $p$ neurons are the input or output ports, and $\circ$ is the element-wise multiplication which picks these neurons out when processing the signals received or sent by the CMU. Note that we do not set the strict limit that one neuron can only work as one role in a CMU, which means the input port and output port may overlap on some neurons. This can actually help the signal transmit faster between CMUs without a direct edge between them.

## 3    ADAPTATION OF CIRCUITNET IN SPECIFIC APPLICATIONS

As a generic network architecture, we expect that CircuitNet can be adapted to all kinds of tasks just like the MLP does. In order to use CircuitNet as a general function approximator, different portions

of the input data should be fed into different Circuit Motif Units as initialization. Ideally, each CMU is responsible for a specific sub-task. After that, CircuitNet will perform several iterations of signal transmission as aforementioned, and a task-specific aggregation transformation will finally extract the information from the network and make a prediction.

The general structure of CircuitNet is quite different from traditional ANNs although they both take the neurons as the most basic building blocks, so that some commonly-used techniques may or may not be applied directly in CircuitNet. Before diving into the details of the experiments, in this section, we briefly show typical cases of how CircuitNet can be adapted in specific tasks. More details on the designs of our model under different circumstances are included in Appendix B

## 3.1 Deep Reinforcement Learning Tasks

One of the tasks we choose to evaluate CircuitNet is deep reinforcement learning (DRL). Here, we choose a range of challenging continuous control tasks with proprioceptive observations (e.g., positions & velocities), which have been widely used as the benchmarks in DRL research (Schulman et al., 2015; 2017; Haarnoja et al., 2018). In these tasks, MLPs are usually used as backbones for policy and value networks. Thus, we directly substitute MLPs with CircuitNet without changing other parts in the learning process.

## 3.2 Image Classification

Another chosen task is image classification in computer vision to investigate how CircuitNet behaves in large-scale structured data. Individual CMUs in CircuitNet are designed to work for specific tasks or embed specific meanings. In computer vision tasks, it is natural to use CMUs to represent features in different locations. Specifically, CircuitNet will start the signal transmission after a convolutional layer initializes neurons states in grid-arranged CMUs with representations of image regions.

Previous MLPs work well on small datasets, but can hardly scale up due to the gradient vanishing problem (Nair & Hinton, 2010). Therefore, many advanced techniques have been proposed to help ANNs to work with deeper networks (He et al., 2016; Simonyan & Zisserman, 2014). Correspondingly, we also adopt similar ideas to help CircuitNet perform better when the numbers of CMUs and iterations of signal transformation are large including weight sharing, normalization, and feature aggregation. Also, to stabilize the training when we stack the quadratic terms over a large number of iterations, we propose a novel **clipped sinusoidal activation function** $\text{ClipSin}(x)$, as follows:

$$\text{ClipSin}(x) = \begin{cases} -A + \epsilon(x+3), & x < -3, \\ A\sin(\frac{\pi}{6}x), & -3 \leq x \leq 3, \\ A + \epsilon(x-3), & x > 3. \end{cases} \tag{8}$$

The activation function is half a period of a sinusoidal function within [-3, 3], and is clipped to a linear function outside with small slope to prevent zero gradients. The amplitude is controlled by a hyperparameter $A$. Practically, we set $\epsilon = 0.01$ and $A = 3.388235$. In this way, we can avoid the gradient exploding problem caused by ReLU activation or the gradient vanishing problem caused by Tanh activation when optimizing deep CircuitNet models. A theoretical justification on why we suppose it to work and how we choose the hyperparameters can be found in Appendix C.

## 3.3 Time Series Forecasting

As a general neural network architecture, CircuitNet can also be applied to sequence data in an autoregressive manner just like the Recurrent Neural Network (RNN) and its variants (Hochreiter & Schmidhuber, 1997; Cho et al., 2014). To demonstrate how this can be achieved, we adopt CircuitNet to several time series forecasting tasks.

The modification of CircuitNet to an autoregressive model is simple and straightforward. First, during every iteration, the state of a specific CMU will be updated with both the signals from all the CMUs and the information from the input at this timestep. Second, since the number of iterations to transmit signals may vary according to the lengths of the input, the parameters are shared across iterations. In this way, we rewrite Eq. (6) to an auto-regressive form as:

$$\mathbf{X_i}^{(k)} = \sigma(g_i(\mathbf{X_i}^{(k-1)}; \mathbf{X_i}^{(k-1)}) + \sum_j^{e_{ji} \in \mathcal{E}} h_{ij}(\mathbf{X_j}^{(k-1)}) + e_i(t_k) + \mathbf{b_i}). \tag{9}$$

$$e_i(t_k) = \sigma(\mathbf{w_{in,i}}(t_k) + \mathbf{b_{in,i}}).$$ (10)

In this form of the state update equation, $\mathbf{w_{in,i}}$ and $\mathbf{b_{in,i}}$ are the iteration-invariant learnable parameters to map the $k$-th input in the sequence to the $i$-th group. After consuming the whole sequence, the final states of the CMUs are used to predict the following data points by a linear forecasting layer. We call the CircuitNet in this form **Recurrent CircuitNet (R-CircuitNet)** to distinguish with the non-autoregressive version introduced in Section 2.

## 4 EXPERIMENTS AND ANALYSIS

In this section, we present the experiment results and analysis on several tasks. Introductions to the datasets and the metrics used to compare CircuitNet with baselines can be found in the Appendix A. Although our primary goal is not to beat the state-of-the-art models, we list them as a comparison. A further study on the limitations and possible improvements are presented in the Appendix D.3

### 4.1 SYNTHETIC EXPERIMENTS

The universal approximation theorem states that a feed-forward network with bounded depth can be an arbitrarily good approximator for any continuous functions if the width is allowed to grow (Hornik et al., 1989; Scarselli & Tsoi, 1998). However, in practice, it may take a lot of neurons and parameters to fit a rather simple function such as the multiplication between two numbers, if linear transformation is the only allowed computation before activation function.

Because the Circuit Motif Units are able to model universal circuit motifs, we expect our network can be used to better approximate some complex functions with equal number of neurons or parameters than MLP. Therefore, we design four types of functions and generated the synthetic datasets, upon which we conduct the experiments to demonstrate the function approximation ability of CMU. Appendix A.1 includes details of the synthetic functions.

#### 4.1.1 RESULTS ON SYNTHETIC DATASETS

The performance of the two different models on the synthetic datasets is presented in Table 1. Models under direct comparison for the same dataset have similar numbers of parameters. From the table, we can see that CMU obtains a significantly lower error than MLP across all the functions, indicating the better ability of CMU to perform universal function approximation by modeling the complex interactions between signals.

Table 1: Fitting errors on the synthetic datasets. $n$ denotes the number of neurons.

| Univariate Exponential | | Univariate Periodic | | Multivariate Linear | | Multivariate Quadratic | |
|---|---|---|---|---|---|---|---|
| **Model** | **RRSE** | **Model** | **RRSE** | **Model** | **RRSE** | **Model** | **RRSE** |
| MLP (n=7) | 0.0021 | MLP (n=64) | 0.0057 | MLP (n=60) | 0.0029 | MLP (n=129) | 0.0077 |
| CMU (n=3) | **0.0013** | CMU (n=15) | **0.0027** | CMU (n=15) | **0.0010** | CMU (n=25) | **0.0062** |

#### 4.1.2 VISUALIZATION OF THE LOSS LANDSCAPES

To gain a deeper insight into the optimization process, we plot in Figure 3 the loss landscapes of MLP and CMU on the two multivariate datasets with GradVis (Chatzimichailidis et al., 2019), in which the parameter space is projected onto the plane formed by the eigenvectors corresponding to the largest two eigenvalues of the Hessian matrix.

From the figure, we can see that the landscapes are rather smooth for MLP and CMU. On the multivariate linear dataset, both models start to converge properly to the minimal point within a few epochs. However, the behavior of CMU differs with MLP on multivariate quadratic dataset. While CMU is still able to find the right direction to converge within 10 epochs, MLP drifts for over 50 epochs before the final converging process. Around these points, the eigenvalues of the Hessian matrix are close to zero or negative and the MLP model struggles to escape from the relatively flat saddle points (Dauphin et al., 2014; Chatzimichailidis et al., 2019). This suggests that by modeling different kinds of motifs, CMU can not only approximate the function more accurately, but also shape the loss landscape advantageously for a quicker convergence.

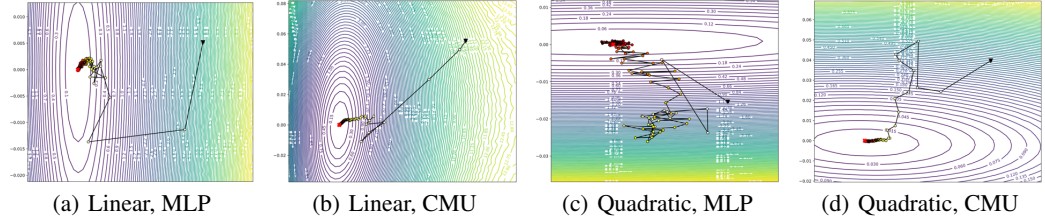

| (a) Linear, MLP | (b) Linear, CMU | (c) Quadratic, MLP | (d) Quadratic, CMU |

Figure 3: Loss landscape and the optimization trajectory of MLP and CMU in the first 100 epochs.

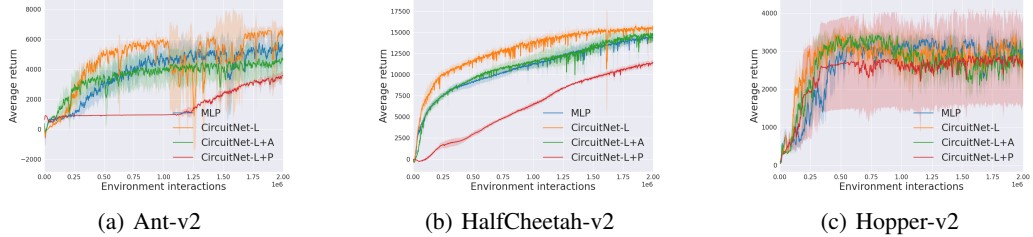

| (a) Ant-v2 | (b) HalfCheetah-v2 | (c) Hopper-v2 |

Figure 4: Learning curves of CircuitNet and MLP on the first three continuous control benchmarks. Appendix C.4 contains the other two without any influence to the conclusion.

## 4.2 DEEP REINFORCEMENT LEARNING

The goal of DRL algorithms is to achieve a lower sample complexity or higher sample efficiency, i.e., to obtain a near-optimal policy using less interactions with the environments. We compare CircuitNet with MLP as the backbones of the Soft Actor-Critic (SAC) (Haarnoja et al., 2018) policy on several challenging continuous control tasks from the OpenAI gym benchmark suite (Brockman et al., 2016). Besides, to compare between different kinds of transformation methods, we run the SAC policy with several variants of our models. Refer to Appendix B.2 for more training details.

The learning curves on these continuous control benchmarks are shown in Figure 4. L, P, A after CircuitNet represents the 3 types of transformations (linear transformation, product between neurons, neuron-wise attention) respectively. Among the shown 3 environments here, CircuitNet achieves significantly better performance than MLP on 2 benchmarks and is comparable on the third environment. It's notable that with product between neurons, CircuitNet fails in DRL tasks. This may be because the overly small number of neurons in CMU (in order to achieve comparable number of parameters) conversely harms the expressiveness of the network.

## 4.3 IMAGE CLASSIFICATION

We conduct experiments on the image classification tasks on MNIST(LeCun & Cortes, 2010), CIFAR-10/100(Krizhevsky et al., 2009) and ImageNet Deng et al. (2009) to demonstrate how CircuitNet performs after scaling up with techniques listed in Section 3.2.

Table 2: Experiment results on MNIST.

| Model | Accuracy | # Params (M) |
|---|---|---|
| 3-layer MLP, 500 + 150 hidden units (LeCun et al., 1998) | 0.976 | 0.47 |
| 2-layer MLP, 800 hidden units (Simard et al., 2003) | 0.984 | 0.64 |
| CircuitNet | **0.993** | 0.14 |
| Ensemble of ensembles(An et al., 2020) | 0.999 | - |

**MNIST** We design a tiny CircuitNet containing 64 neurons and 48 ports in every CMU and for every batch of inputs with two iterations of signal transmissions. As shown in Table 2, comparing with two MLP models, CircuitNet achieves a higher accuracy with less than 30% of parameters.

**CIFAR-10/100** For CIFAR datasets, we design a larger CircuitNet containing 32 neurons and 24 ports in each CMU with 9 iterations of signal transmissions. As shown in Table 3, we com-

Table 3: Experiment results on CIFAR-10 and CIFAR-100.

| Model | Accuracy | | # Params (M) | # FLOPs (M) |
|---|---|---|---|---|
| | CIFAR-10 | CIFAR-100 | | |
| 12-layer MLP | 0.572 | 0.246 | 5.2 | 5.5 |
| ResNet-18 (He et al., 2016) | 0.870 | 0.570 | 11.2 | 75.1 |
| VIT (Dosovitskiy et al., 2020) | 0.809 | 0.542 | 9.6 | 1283 |
| Swin Transformer (Liu et al., 2021) | 0.846 | 0.554 | 26.7 | 1049 |
| MLP-Mixer (Tolstikhin et al., 2021) | 0.801 | 0.547 | 18.7 | 406 |
| CircuitNet-L | 0.858 | 0.588 | 15.5 | 39.7 |
| CircuitNet-L+A | 0.853 | 0.588 | 15.7 | 74.9 |
| CircuitNet-L+P | **0.913** | **0.649** | 4.0 | 41.3 |
| Efficient adaptive ensembling(Bruno et al., 2022) | 0.996 | 0.968 | - | - |

Table 4: Experiment results on ImageNet. The baseline results were from (Han et al., 2022).

| Model | # Params (M) | # FLOPs (B) | Accuracy |
|---|---|---|---|
| ResNet-18 (He et al., 2016) | 12 | 1.8 | 0.706 |
| ViT-B/16 (Dosovitskiy et al., 2020) | 86.4 | 55.5 | 0.779 |
| PVT-Tiny (Wang et al., 2021) | 13.2 | 1.9 | 0.751 |
| ResMLP-S12 (Touvron et al., 2022) | 16.7 | 3.2 | 0.770 |
| Poolformer-S12 (Yu et al., 2022b) | 12 | 2.0 | 0.772 |
| Pyramid ViG-Ti (Han et al., 2022) | 10.7 | 1.7 | 0.782 |
| CircuitNet | 11.0 | 1.8 | **0.785** |
| CoCa(Yu et al., 2022a) | 2100 | - | 0.910 |

pare CircuitNet with some of the existing SOTA image classification models, including CNN-based method (ResNet), transformer-based methods (ViT and Swin Transformer) and MLP-based method (MLP-Mixer). Since CIFAR-10/100 are relatively small, transformer-based methods achieve worse performance due to lack of training data, e.g., without pretraining on ImageNet. CircuitNet outperforms all the compared methods when applying product to model the interactions between neurons. Meanwhile, by doubling number of neurons and ports, CircuitNet achieves comparable performance with only linear or neuron-wise attention (i.e., CircuitNet-L and CircuitNet-L+A).

**ImageNet** As shown in Table 4, by further scaling up CircuitNet to 12 transmission iterations, on the large-scale ImageNet dataset, CircuitNet achieves better performance than existing state-of-the-art methods (with comparable or larger number of parameters) including CNN-based method (ResNet), MLP-based method (ResMLP), transformer-based methods (ViT-B/16, PVT-Tiny, and Poolformer-S12) and a very recent GNN-based method (Pyramid ViG-Ti).

## 4.4 TIME SERIES FORECASTING

To test the ability of CircuitNet in the autoregressive manner, i.e., Recurrent CircuitNet (R-CircuitNet), to be applied to sequential data, we include several multivariate time series forecasting tasks and compare our model to various kinds of time series forecasting methods.

The time series forecasting results are presented in Table 5. Statistical methods (ARIMA (Box et al., 2015), GP (Roberts et al., 2013)), CNN-based model (TCN (Lea et al., 2017)), RNN-based model (GRU (Cho et al., 2014)), transformer based-models (Autoformer (Wu et al., 2021), Informer (Zhou et al., 2021)), and hybrid model (LSTNet (Lai et al., 2018)) are all listed for comparison and they tend to be effective in one or two specific settings. As for R-CircuitNet, the three different variants perform similarly and outperform strong baselines by a considerable margin across half of the tasks, and can still get comparable numbers in others.

## 5 RELATED WORKS

Many previous neural network architectures were designed by closely mimicking biological neural organizations (Bengio et al., 2015). One of the earliest prior arts might be single/multi-layer perceptron (Rosenblatt, 1958; Minsky & Papert, 1969), which was a mathematical simplification of signals

Table 5: Experiment results on time series datasets with different forecasting horizons (6, 24).

| Model | Solar-energy | | | | Pems-bay | | | | Metr-la | | | |
| | 6 | | 24 | | 6 | | 24 | | 6 | | 24 | |
| | RRSE↓ | R2↑ | RRSE↓ | R2↑ | RRSE↓ | R2↑ | RRSE↓ | R2↑ | RRSE↓ | R2↑ | RRSE↓ | R2↑ |
|---|---|---|---|---|---|---|---|---|---|---|---|---|
| ARIMA (Box et al., 2015) | **0.202** | 0.951 | 0.365 | 0.847 | 0.532 | 0.741 | 0.548 | 0.723 | 0.575 | 0.687 | 0.742 | 0.441 |
| GP (Roberts et al., 2013) | 0.225 | 0.944 | 0.388 | 0.836 | 0.544 | 0.732 | **0.532** | 0.712 | 0.572 | 0.685 | 0.738 | 0.437 |
| LSTNet (Lai et al., 2018) | 0.218 | 0.950 | 0.360 | 0.857 | 0.521 | 0.752 | 0.560 | 0.706 | 0.564 | 0.697 | 0.732 | 0.475 |
| GRU (Cho et al., 2014) | 0.219 | 0.950 | **0.355** | 0.875 | 0.529 | 0.747 | 0.573 | 0.703 | 0.517 | 0.759 | 0.797 | 0.429 |
| TCN (Lea et al., 2017) | 0.210 | 0.954 | 0.445 | 0.804 | 0.487 | 0.785 | 0.551 | 0.718 | 0.570 | 0.698 | 0.705 | 0.545 |
| Informer (Zhou et al., 2021) | 0.239 | 0.948 | 0.358 | **0.889** | 0.522 | 0.755 | 0.547 | **0.733** | 0.579 | 0.697 | 0.747 | 0.516 |
| R-CircuitNet-L | 0.216 | 0.956 | 0.382 | 0.861 | 0.378 | 0.869 | 0.545 | 0.730 | 0.450 | 0.818 | 0.692 | 0.570 |
| R-CircuitNet-L+A | 0.212 | 0.957 | 0.409 | 0.841 | **0.377** | **0.870** | 0.561 | 0.712 | 0.448 | **0.820** | 0.667 | 0.600 |
| R-CircuitNet-L+P | 0.211 | **0.958** | 0.515 | 0.748 | 0.380 | 0.868 | 0.563 | 0.711 | **0.447** | 0.820 | **0.656** | **0.613** |
| Autoformer (Wu et al., 2021) | 0.212 | 0.960 | 0.432 | 0.852 | 0.452 | 0.782 | 0.543 | 0.711 | 0.565 | 0.762 | 0.692 | 0.548 |

being received from dendrites and sent down to the axon once activated. Convolutional neural networks (Fukushima & Miyake, 1982; LeCun et al., 1998; Krizhevsky et al., 2012) apply filters over an input image, which can learn feature maps akin to the planes of S-cells in the Neocognitron (Hubel & Wiesel, 1968; Fukushima & Miyake, 1982; Lindsay, 2021). Recurrent neural networks (Hochreiter & Schmidhuber, 1997; Cho et al., 2014) achieved great successes in modeling sequential dependencies among the data due to mimicking the recurrent connections within and between stages of the cascade in the human brain (Kietzmann et al., 2019; van Bergen & Kriegeskorte, 2020). In spiking neural networks (Maass, 1997; Woźniak et al., 2020), each neuron fires when its input crosses a threshold, which was inspired by the occurrence of information transmission when a membrane potential reaches a threshold in the cortex (Gerstner & Kistler, 2002; Ghosh-Dastidar & Adeli, 2009). Inspired by visual information representation in human visual cortex (Qiao et al., 2018), the capsule neural network adds structures called "capsules" to a CNN, each of which represents an object or object part and the combinations of which form more stable representations. Although the above neural network architectures have achieved huge successes in specific tasks, they are not generic enough to model the combinations of different types of neural circuits, e.g., a combination of feedforward excitation/inhibition, lateral inhibition and mutual inhibition in human brains (Luo, 2021).

Besides neural network architectures, the successes of many popular operations in deep learning can also be explained from the perspective of biological neural processes. For instance, attention mechanism has become an integral part in many sequence modeling algorithms, which can capture both local dependencies and long-range dependencies in the sequence or space (Bahdanau et al., 2014; Vaswani et al., 2017; Veličković et al., 2018). In the biological visual systems, attention mechanism is also considered as an important ability to flexibly control with limited computational resources in nervous systems, in which neurons that represent the attended features have higher firing rates than those represent unattended features (Treue & Trujillo, 1999; Lindsay, 2020). Dropout (Srivastava et al., 2014) is another popular technique for preventing deep neural networks from overfitting, which is analogy to the characteristic of biological neural networks randomly switching neurons on or off during training and inference (Faisal et al., 2008; Deco et al., 2009). Normalizations, such as batch normalization (Ioffe & Szegedy, 2015) and weight normalization (Salimans & Kingma, 2016), are popular techniques to facilitate the training of deep neural networks. This phenomenon has been found from invertebrates to mammals as a canonical neural computation, in which the responses of some neurons are divided by a common factor reflecting the summed activity of a pool of neurons (Carandini & Heeger, 2012). The above works are orthogonal to the proposed CircuitNet, and we leverage these operations to improve the performance of CircuitNet in machine learning tasks.

## 6 CONCLUSION

This paper proposes a new type of neural network inspired by the architecture of neural circuits, where groups of densely connected neurons, namely Circuit Motif Units, form the basic units for modeling universal circuit motifs and a locally-dense and globally-sparse high-level network structure is applied inspired by the connections among functional regions of brain. Experiments on synthetic and real datasets demonstrate that CircuitNet can outperform popular neural network architectures, such as MLP, CNN, RNN, and transformer, in several kinds of machine learning tasks.

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

# A    DETAILS OF THE DATASETS

## A.1    SYNTHETIC DATASETS

The synthetic datasets are generated via:

$$y = f(\mathbf{X}),$$

where $f(\cdot)$ is a known function which the model needs to fit. We include both univariate and multivariate functions in our synthetic dataset.

- The univariate functions are from Zainuddin & Pauline (2008) with proper normalization. The train set and test set are generated by evenly sampling $x$ within [-1, 1].
- The multivariate functions are generated by sampling randomly $a_k$ from $\mathcal{U}(-1, 1)$ and $i_k$, $j_k$ from $\{1, 2, ..., 10\}$. The train set and test set are generated by randomly sampling $X_i$ from $\mathcal{U}(-1, 1)$.

The details of the datasets are presented in Table 6. We leverage the Root Relative Squared Error (RRSE) to evaluate the fitting quality:

$$\text{RRSE} = \sqrt{\frac{\sum_{m=1}^{M} \|y_m - \hat{y}_m\|^2}{\sum_{m=1}^{M} \|y_m - \bar{y}\|^2}}, \tag{11}$$

where $M$ is the size of the test set and $\bar{y}$ is the average of $y_m$.

Table 6: Synthetic datasets

| Dataset Name | $y = f(x)$ or $f(\mathbf{X})$ | # Train | # Test | Dimension of X | Range of $X_i$ |
|---|---|---|---|---|---|
| Univariate Exponential | $(x+1)e^{-3x+3}/50$ | 100 | 200 | 1 | [-1, 1] |
| Univariate Periodic | $\sin(4\pi x)e^{-5x}$ | 100 | 200 | 1 | [-1, 1] |
| Multivariate Linear | $\sum_{k=1}^{20} a_k X_k$ | 2,000 | 200 | 20 | [-1, 1] |
| Multivariate Quadratic | $\sum_{k=1}^{20} a_k X_{i_k} X_{j_k}$ | 2,000 | 200 | 10 | [-1, 1] |

## A.2    REINFORCEMENT LEARNING FOR CONTINUOUS CONTROL

The continuous control experiment on 5 different benchmarks are fetched directly from the OpenAI gym benchmark suite (Brockman et al., 2016). The dimensions of the observation space and action space are listed in Table 7:

## A.3    IMAGE CLASSIFICATION

CircuitNet is evaluated on three public image classification datasets, namely MNIST (LeCun & Cortes, 2010), CIFAR(Krizhevsky et al., 2009) and ImageNet (Deng et al., 2009). MNIST is a handwriting digit recognition dataset using 60,000 images as training set and 10,000 images as test set, with ten categories, 0 to 9. The CIFAR-10 dataset consists of 50,000 images of size $32 \times 32$ in

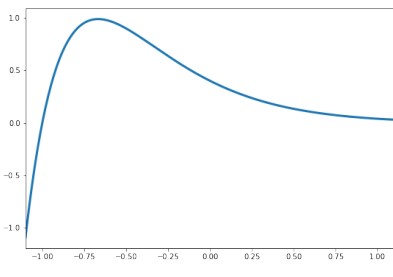

(a) Univariate Exponential

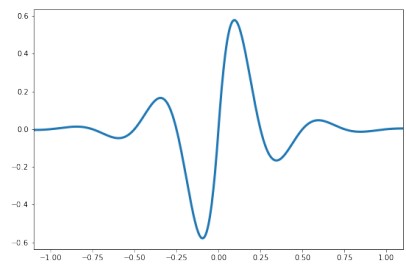

(b) Univariate Periodic

Figure 5: The two univariate functions in synthetic datasets.

Table 7: Continuous control environments.

| Environment | Observation dimension | Action dimension |
|---|---|---|
| Ant-v2 | 111 | 8 |
| HalfCheetah-v2 | 17 | 6 |
| Hopper-v2 | 11 | 3 |
| Humanoid-v2 | 376 | 17 |
| Walker2d-v2 | 17 | 6 |

training set and 10,000 images in test set with 10 classes. CIFAR100 has the same train/test split but contains 100 classes in total. For ImageNet dataset we use the most common ImageNet (ILSVRC) subset that contains 1,000 object classes with totally 1,281,167 training images and 50,000 validation images.

### A.4 TIME SERIES FORECASTING

We have leveraged three common benchmarking datasets for time series forecasting task. Solar (Lai et al., 2018) includes the solar power production records in 2006, which are sampled every 10 minutes from 137 PV plants in Alabama State. Pems-bay (Li et al., 2017) contains average traffic speed records measured by 325 sensors in the Bay Area ranging from Jan 2017 to May 2017, offered by California Transportation Agencies (CalTrans). Metr-la (Li et al., 2017) contains average traffic speed measured by 207 loop detectors on the highways of Los Angeles County ranging from Mar 2012 to Jun 2012, provided by the Los Angeles Metropolitan Transportation Authority.

In addition to the Root Relative Squared Error (RRSE) in Equation (11), we also include coefficient of determination ($R^2$) as the evaluation metric:

$$R^2 = \frac{1}{MNT} \sum_{m=1}^{M} \sum_{i=1}^{N} \sum_{t=1}^{T} \left[ 1 - \frac{\|y_m^{i,t} - y_m^{i,t}\|}{\|y_m^{i,t} - \bar{y}^{i,t}\|} \right]. \tag{12}$$

Here, $M$ is the size of test sets, $N$ is the number of variables, and $T$ is the forecasting horizon. $y_m^{i,t}$ is the $t$-th future value of the $i$-th variable for the $m$-th sample in the test set, and $\bar{y}$ is the average over all test samples.

## B MODEL CHOICES AND TRAINING DETAILS

We list the details of training for reproducing.

### B.1 SYNTHETIC DATASETS

We use an MLP with 3 layers for this experiment. This is the closest structure to our CMU, where input data is first mapped to CMU via a linear function, then one iteration of signal transmission

is conducted with both linear transformation, neuron-wise attention, and product between neurons, and the 1-dimension result is obtained by another linear function. Since modeling universal motifs introduces more parameters, we list models with similar number of parameters rather than hidden neurons for fair comparison in Table 1.

When training with gradient descent optimizer in the two univariate datasets, we find that both models will occasionally stuck in some local minima. Therefore, we leverage the Adam optimizer Kingma & Ba (2015). As for the multivariate functions, because we need to obtain the optimization trajectory for analysis, we still use the mini-batch gradient descent algorithm with batch size of 128. The learning rate was set to $1 \times 10^{-2}$ with 0.9 momentum, and all the models are optimized for 10,000 epochs at maximum to ensure convergence.

## B.2 Reinforcement Learning for Continuous Control

In this paper, we use Soft Actor-Critic (SAC) (Haarnoja et al., 2018), a popular off-policy deep reinforcement learning (DRL) algorithm with high sample efficiency as an example, but our method can be similarly applied to other Reinforcement Learning algorithms using neural networks as value function approximators or policy networks. We use CircuitNet as backbones for the policy function and two Q-functions in the SAC algorithm. Intuitively, for tasks where the states and actions have specific and known meanings, we may divide the inputs into groups based on their meanings and each group corresponds to the input of a CMU in CircuitNet. However, the physical meaning of the states and actions are not exposed in selected tasks, so we simply map all the input dimensions to groups with differently initialized linear layers. The CircuitNets used in DRL tasks are relatively small and do not have many CMUs, so the connections among CMUs can be designed as a complete graph without introducing too many parameters.

The detailed hyperparameters are listed in Table 8. The sizes of the network are designed to match the number of parameters of the MLP baseline in the original paper. As for the learning rate and the activation function, we conducted grid search for every possible value in the sets and the best results of each model is reported. Because of the high fluctuation in deep reinforcement learning, we ran all the experiments with 4 different seeds to present the error bar in the figure.

Table 8: Training details for deep reinforcement learning

| Soft Actor Critic | |
|---|---|
| Optimizer | Mini-batch gradient descent |
| Batch size | 256 |
| Learning rate | $\{3 \times 10^{-4}, 1 \times 10^{-4}, 5 \times 10^{-5}, 3 \times 10^{-5}, 1 \times 10^{-4}\}$ |
| Discount | 0.99 |
| Starting Steps | 10000 |
| Replay buffer size | $1 \times 10^{6}$ |
| **Model Details** | |
| # CMUs | 4 |
| # Iterations of signal transmission | 2 |
| # Neurons per CMU | |
|    CircuitNet-L | 64 |
|    CircuitNet-L+A | 40 |
|    CircuitNet-L+P | 32 |
| # Ports per CMU | |
|    CircuitNet-L | 48 |
|    CircuitNet-L+A | 30 |
|    CircuitNet-L+P | 24 |
| Activation function | {ReLU, ClipSin} |

## B.3 Image Classification

### B.3.1 General Design Choices

**Topology of CMUs**  We first map the input image to CMUs based on the regions. Specifically, a convolution layer is applied to the input image and extracts representations for different regions. Then we directly use the representations to initialize the states of the neurons in CMUs. As a result, each CMU corresponds to a region of the input image after initialization and the CMUs are arranged as grids.

For MNIST and CIFAR datasets, to transmit signals among CMUs, we use $2^k$-hop connections ($k \in \mathbb{N}$), meaning that the $(i \times 2^k)$-th CMU in every row and every column can communicate with the $[(i-1) \times 2^k]$-th and $[(i+1) \times 2^k]$-th CMUs. This topology keeps the global sparsity with $O(N)$ connections while enabling arbitrary two CMUs to communicate within $O(\log N)$ iterations, where $N$ is the total number of CMUs.

For ImageNet dataset, because of the huge number of CMUs, even grid connections will introduce too many edges. Thus we follow Han et al. (2022) and consider the correlation between CMU signals by adding edges between CMUs with top-$K(K = 9)$ lowest output signal distance to accelerate signal transmission process.

**Weight Sharing**  In order to reduce the dimension of the parameter space and utilize the spatial-invariant property for the image classification task, weights are shared across CMUs in the same iteration to transmit signals within the units. However, since the topological position of CMUs are not completely symmetric, inter-unit weights between ports are not shared.

**Weight Normalization**  Normalization is a useful technique to stabilize the training process. Besides batch normalization, in CircuitNet, we also consider the weight normalization (Salimans & Kingma, 2016) as an alternative option. If we rewrite the update of neurons as:

$$\mathbf{X_i}^{(k)} = \phi(\mathbf{V}\hat{\mathbf{X}}^{(k-1)}), \tag{13}$$

instead of using $\mathbf{V}$ directly, we use its direction $\mathbf{W}$ to do the transformation:

$$\mathbf{W} = \frac{\mathbf{V}}{\|\mathbf{V}\|}. \tag{14}$$

Here, $\hat{\mathbf{X}}^{(k-1)}$ is a vector whose elements can be calculated by the states from the previous iteration $k-1$ and constants and $\mathbf{V}$ is the learnable parameters in this iteration.

We use this type of normalization to address the problem caused by the different number of inputs for different neurons due to the incoming edges and different roles of neurons. Without the weight normalization technique, this problem becomes even harmful when parameters are shared across CMUs which are topologically asymmetric.

**Feature Aggregation**  Compared to the number of categories in image classification datasets, the hidden size in every iteration is still too high to project onto the output space. Therefore, we apply learnable linear aggregation function similar to pooling layers to merge different CMUs together based on the spatial locations after certain iterations.

### B.3.2  Training Details for Different Datasets

**MNIST**  We design a tiny CircuitNet that contains $4 \times 4$ groups with each CMUs containing 64 neurons and 48 ports, and the signal transmission is conducted for two iterations.

**CIFAR**  CircuitNet-L+P contains $16 \times 16$ CMUs with each CMU containing 32 neurons and 24 ports, the signal transmission is conducted for 9 iterations. To reduce the computation cost and accelerate the signal transmission, we merge the neighboring 4 CMUs at the 3rd and 6th signal transmission iteration to reduce the total CMU numbers, which cut the number of CMUs to 1/4 after each merge operation. CircuitNet-L simply leverages Equation (1) to transmit signals and CircuitNet-L+A replace the product transformation in Equation (5) with neuron-wise attention in (2) and (3). In order to make the these two CircuitNet variants to have comparable number of parameters and/or FLOPs, the CMU size is increased to 64, 52 neurons and the port size is increase to 48, 39 ports, respectively.

Table 9: Training details for Image Classification. Hyperparameters in braces are searched.

| Optimization | |
| --- | --- |
| Optimizer | {Adam, SGD} |
| Maximum learning rate | $\{1 \times 10^{-1}, 1 \times 10^{-2}, 1 \times 10^{-3}, 5 \times 10^{-4}, 1 \times 10^{-4}\}$ |
| # Warm-up iterations | {0, 5} |
| Learning rate decay | Cosine Annealing Scheduler |
| Number of epochs | 600 |
| **Model Details** | |
| # CMUs | |
|   MNIST | $4 \times 4$ |
|   CIFAR | $16 \times 16$ |
|   ImageNet | $112 \times 112$ |
| # Iterations of signal transmission | |
|   MNIST | 2 |
|   CIFAR | 9 |
|   ImageNet | 12 |
| # Neurons per CMU | |
|   MNIST | 64 |
|   CIFAR | |
|     CircuitNet-L | 64 |
|     CircuitNet-L+A | 52 |
|     CircuitNet-L+P | 32 |
|   ImageNet | 48 |
| # Ports per CMU | |
|   MNIST | 48 |
|   CIFAR | |
|     CircuitNet-L | 48 |
|     CircuitNet-L+A | 39 |
|     CircuitNet-L+P | 24 |
|   ImageNet | 36 |
| Activation function | {Tanh, ReLU, ClipSin} |
| Normalization | {Batch Normalization, Weight Normalization} |

**ImageNet** We further scale-up the CircuitNet to achieve better performance on ImageNet containing 12 transmission iterations splitted to 4 stages with 2, 2, 4 and 2 iterations, respectively. Following Han et al. (2022), to suppress over-smoothing, a linear transformation is added both before and after the signal transmission. Furthermore, before feed inputs into the CircuitNet, we add a Stem structure containing 3 convolution layers that down-sample the original image to a $112 \times 112$ feature-map. Each feature vector in the feature-map corresponds to a initial state of a CMU.

Currently, for ImageNet, CircutNet applies only linear transformation for intra-CMU signal transmission but already achieves comparable performance with existing state-of-the-art methods.

For training implementation, random-crop and horizontal-flip data augmentations are applied. The hyperparameters are summarized in Table 9.

### B.4 TIME SERIES FORECASTING

The recurrent CircuitNet initialize all the states of CMUs with zeros at the beginning, i.e., $\mathbf{X_i}^{(0)} = \mathbf{0}$. Then it consumes the incoming data point, maps it to CMUs, and updates the states according to Equation (9). After the whole time series are proceeded, we obtained the feature vector of the sequence by down-sampling from the final states of all the CMUs.

The tasks introduced in A.4 are multivariate time series forecasting tasks. However, in order to limit the extra designs to the minimal, we simplify the multivariate inputs to univariate time series.

Specifically, we use the same recurrent CircuitNet to generate the feature vectors for all the variate sequences separately, and aggregate them before the forecasting layer.

In our experiment, we have $1 \sim 12$ CMUs for the recurrent CircuitNet. Thus the connections among CMUs are designed as a complete graph similar to the case in reinforcement learning. A summary of hyperparameters are included in Table 10. We run all the experiments with 3 different seeds and the average results are reported.

Table 10: Training details for time series forecasting. Hyperparameters in braces are searched.

| Optimization | |
| --- | --- |
| Optimizer | Adam |
| Loss function | MSELoss |
| Learning rate | $1 \times 10^{-3}$ |
| Weight decay | $1 \times 10^{-5}$ |
| Number of epochs | 1000 |
| **Model Details** | |
| # Neurons per CMU | $\{4, 8, 16\}$ |
| # Ports per CMU | $0.75 \times$ # Neurons per CMU |
| # Total Neurons | $\{16, 32, 48\}$ |
| Dimension of feature vector per variate | $\{1, 2, 4, 8\}$ |
| Activation function | Tanh |

## C  ACTIVATION DESIGN FOR QUADRATIC TERMS IN CMU

In the image classification task, the signals will transmit for several iterations within a CMU. When the number of iterations becomes large, it may cause the gradient vanishing or gradient exploding problem, prohibits the model to converge correctly. If all the operations are linear, this problem has already been solved effectively with Rectified Linear Units (ReLU)(Krizhevsky et al., 2012). However, when quadratic terms are introduced in Equation (4) and (5), we need to seek another solution to address this problem. As mentioned in Section 3.2, instead of traditional activation functions, we use a clipped sinusoidal activation function $\text{ClipSin}(x)$:

$$\text{ClipSin}(x) = \begin{cases} -A + \epsilon(x + 3), & x < -3, \\ A\sin(\frac{\pi}{6}x), & -3 \leq x \leq 3, \\ A + \epsilon(x - 3), & x > 3. \end{cases} \tag{15}$$

And we set $\epsilon = 0.01$ and $A = 3.388235$. We provide some theoretical justification on this design here. We will start by investigating how ReLU may prevent the gradient from vanishing or exploding and why it fails in quadratic situations. Then we introduce how to find a satisfactory function and why the proposed $\text{ClipSin}$ may be useful. Finally, we show how we determine the parameters.

### C.1  ReLU ACTIVATION IN LINEAR TRANSMISSIONS

The linear transmission of signals in iteration $k$ with an activation function can be written as:

$$\mathbf{X}^{(k+1)} = \sigma(\mathbf{W}^{(k)}\mathbf{X}^{(k)} + \mathbf{b}^{(k)}), \tag{16}$$

or if we focus on a specific neuron:

$$x_m^{(k+1)} = \sigma(\sum_i w_{mi}^{(k)}x_i^{(k)} + b_m^{(k)}) = \sigma(x_m'^{(k+1)}). \tag{17}$$

The derivative of neurons between iterations is:

$$\frac{\partial x_m^{(k+1)}}{\partial x_n^{(k)}} = \sigma'(x_m'^{(k+1)})w_{mn}^{(k)}. \tag{18}$$

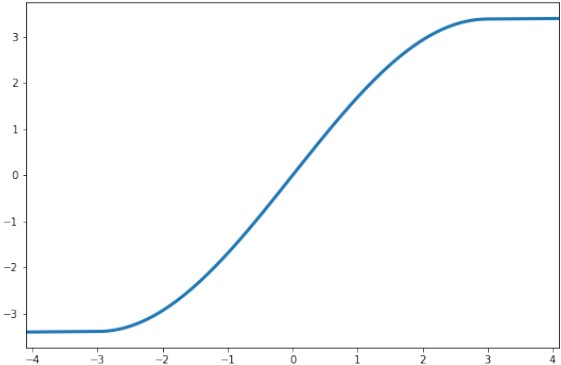

Figure 6: Plot of $y = \text{ClipSin}(x)$

According to the chain rule, the derivatives will be multiplied together with some additional terms introduced by normalization and the optimization target. When the activation function $\sigma(\cdot)$ is the ReLU function, $\sigma'(\cdot)$ is either 0 or 1. This means the ReLU function is essentially *turning on or off* the paths to pass the gradient without bringing in any bias on the gradient scales.

### C.2 GRADIENTS IN QUADRATIC TRANSMISSIONS

When we have the linear and quadratic terms at the same time, the equation of transmission will be rewritten as:

$$\mathbf{X}^{(k+1)} = \sigma(\frac{1}{2}\mathbf{W}^{(k)}\hat{\mathbf{X}}^{(k)} + \mathbf{V}^{(k)}\mathbf{X}^{(k)} + \mathbf{b}^{(k)}), \tag{19}$$

$$x_m^{(k+1)} = \sigma(\frac{1}{2}\sum_{i,j} w_{mij}^{(k)} x_i^{(k)} x_j^{(k)} + \sum_i v_{mi}^{(k)} x_i^{(k)} + b_m^{(k)}) = \sigma(x_m'^{(k+1)}). \tag{20}$$

Here, $\hat{\mathbf{X}}$ is the vector containing the product between every two elements in $\mathbf{X} = (x_1, x_2, ..., x_n)^\top \in \mathbb{R}^n$, i.e. $\hat{\mathbf{X}} = (x_1^2, x_1 x_2, ..., x_n x_{n-1}, x_n^2)^\top \in \mathbb{R}^{n^2}$. Now the derivative of neurons between adjacent iterations is:

$$\frac{\partial x_m^{(k+1)}}{\partial x_n^{(k)}} = \sigma'(x_m'^{(k+1)}) \left[ \frac{1}{2}\sum_{i\neq n}(w_{min}^{(k)} + w_{mni}^{(k)})x_i^{(k)} + w_{mnn}^{(k)}x_n^{(k)} + v_{mn}^{(k)} \right]$$

$$= \sigma'(x_m'^{(k+1)}) \left[ \frac{1}{2}\sum_{i\neq n}(w_{min}^{(k)} + w_{mni}^{(k)})\sigma(x_i'^{(k)}) + w_{mnn}^{(k)}\sigma(x_n'^{(k)}) + v_{mn}^{(k)} \right]$$

$$:= \sigma'(x_m'^{(k+1)}) \cdot t_{mn}^{(k)}. \tag{21}$$

If we apply the chain rule once, we get:

$$\frac{\partial x_m^{(k+1)}}{\partial x_n^{(k-1)}} = \sum_l \frac{\partial x_m^{(k+1)}}{\partial x_l^{(k)}} \frac{\partial x_l^{(k)}}{\partial x_n^{(k-1)}}$$

$$= \sigma'(x_m'^{(k+1)}) \sum_l t_{ml}^{(k)} \sigma'(x_l'^{(k)}) t_{ln}^{(k-1)}$$

$$= \sigma'(x_m'^{(k+1)}) \sum_l t_{ln}^{(k-1)} \cdot \left[ \frac{1}{2}\sum_{i\neq l}(w_{mil}^{(k)} + w_{mli}^{(k)})\sigma(x_i'^{(k)})\sigma'(x_l'^{(k)}) \right.$$

$$\left. + w_{mll}^{(k)}\sigma(x_l'^{(k)})\sigma'(x_l'^{(k)}) + v_{ml}^{(k)}\sigma'(x_l'^{(k)}) \right] \tag{22}$$

From Equation (22), we can find 3 factors from the activation function that influence the gradient scale: $\sigma(x_i'^{(k)})\sigma'(x_l'^{(k)})$, $\sigma(x_l'^{(k)})\sigma'(x_l'^{(k)})$, and $\sigma'(x_l'^{(k)})$, and similar terms will be multiplied when we apply the chain rule to more iterations. Then we can analyze how different activation functions behave:

- $\sigma(x) = \text{ReLU}(x)$. When $x < 0$, $\sigma'(x) = 0$ and it turns off the gradient propagation path properly. However, when $x \geq 0, y \geq 0$, even if we have $\sigma'(x) = \sigma'(y) = 1$, $\sigma(x)\sigma'(y) = \sigma(x)\sigma'(x) = x$ are both unbounded. This will cause gradient explosion or gradient vanishing based on the input scale.

- $\sigma(x) = \tanh(x)$, then we have $|\sigma(x)| < 1$, $|\sigma'(x)| \leq 1$, and $|\sigma(x)\sigma'(x)| \leq \frac{2\sqrt{3}}{9} \approx 0.385$. The occurrence of gradient vanishing is doomed.

Therefore, we must come up with a new activation function which can properly locate all $|\sigma'(x)|$, $|\sigma(y)\sigma'(x)|$, and $|\sigma(x)\sigma'(x)|$ around 1.

### C.3 Clipped Sinusoidal Activation Function

The simplest function we find to be satisfactory is the sinusoidal function $A\sin(\omega x)$. Let $\sigma(x) = A\sin(\omega x)$, we have:

$$|\sigma'(x)| = |A\omega\cos(\omega x)| \leq A\omega, \tag{23}$$

$$|\sigma(y)\sigma'(x)| = |A^2\omega\sin(\omega y)\cos(\omega x)| \leq A^2\omega, \tag{24}$$

$$|\sigma(x)\sigma'(x)| = |\frac{1}{2}A^2\omega\sin(2\omega x)| \leq \frac{1}{2}A^2\omega, \tag{25}$$

which can all be probably bounded and scaled by carefully choosing $A$ and $\omega$.

However, one thing annoys us is the fact that $\sin(x)$ is not monotonic, so we choose to use only half a period of the function in $[-\frac{\pi}{2\omega}, \frac{\pi}{2\omega}]$ and use a linear function with small slope outside this region. The complete format of the Clipped Sinusoidal Activation Function $\text{ClipSin}(x)$ is:

$$\text{ClipSin}(x) = \begin{cases} -A + \epsilon(x + \frac{\pi}{2\omega}), & x < -\frac{\pi}{2\omega}, \\ A\sin(\omega x), & -\frac{\pi}{2\omega} \leq x \leq \frac{\pi}{2\omega}, \\ A + \epsilon(x - \frac{\pi}{2\omega}), & x > \frac{\pi}{2\omega}. \end{cases} \tag{26}$$

### C.4 Choice of Parameters

The intuition of selecting the parameters is to make the scales of the terms in Equation (23), (24), and (25) as close to 1 as possible, or at least, in the sense of expectation. Supposing that we can have a normalization layer before the activation function that normalize $x_l'^{(k)}$ to the standard normal distribution $\mathcal{N}(0,1)$, we first choose $\omega = \frac{\pi}{6}$ so that $[-\frac{\pi}{2\omega}, \frac{\pi}{2\omega}]$, i.e. [-3, 3], contains the majority of $x_l'^{(k)}$. Also, considering that most terms in Equation (22) are $\sigma(x_i'^{(k)})\sigma'(x_l'^{(k)})$, we force:

$$\mathbb{E}_{x_i'^{(k)} \sim \mathcal{N}(0,1), x_l'^{(k)} \sim \mathcal{N}(0,1)}[|A^2\omega\sin(\omega x_i'^{(k)})\cos(\omega x_l'^{(k)})|] = 1. \tag{27}$$

Solving the equation and discarding the negative root, we get:

$$A = 2e^{\frac{\pi^2}{72}}\sqrt{\frac{3}{\pi\text{erfi}(\frac{\pi}{6\sqrt{2}})}} \approx 3.388235, \tag{28}$$

where $\text{erfi}(z) = -i \cdot \text{erf}(iz)$ is the imaginary error function. This is the amplitude of the sinusoidal function we use in our model. As for the slope of the linear function $\epsilon$, we follow the design in LeakyReLU(Maas et al., 2013) to set it to 0.01.

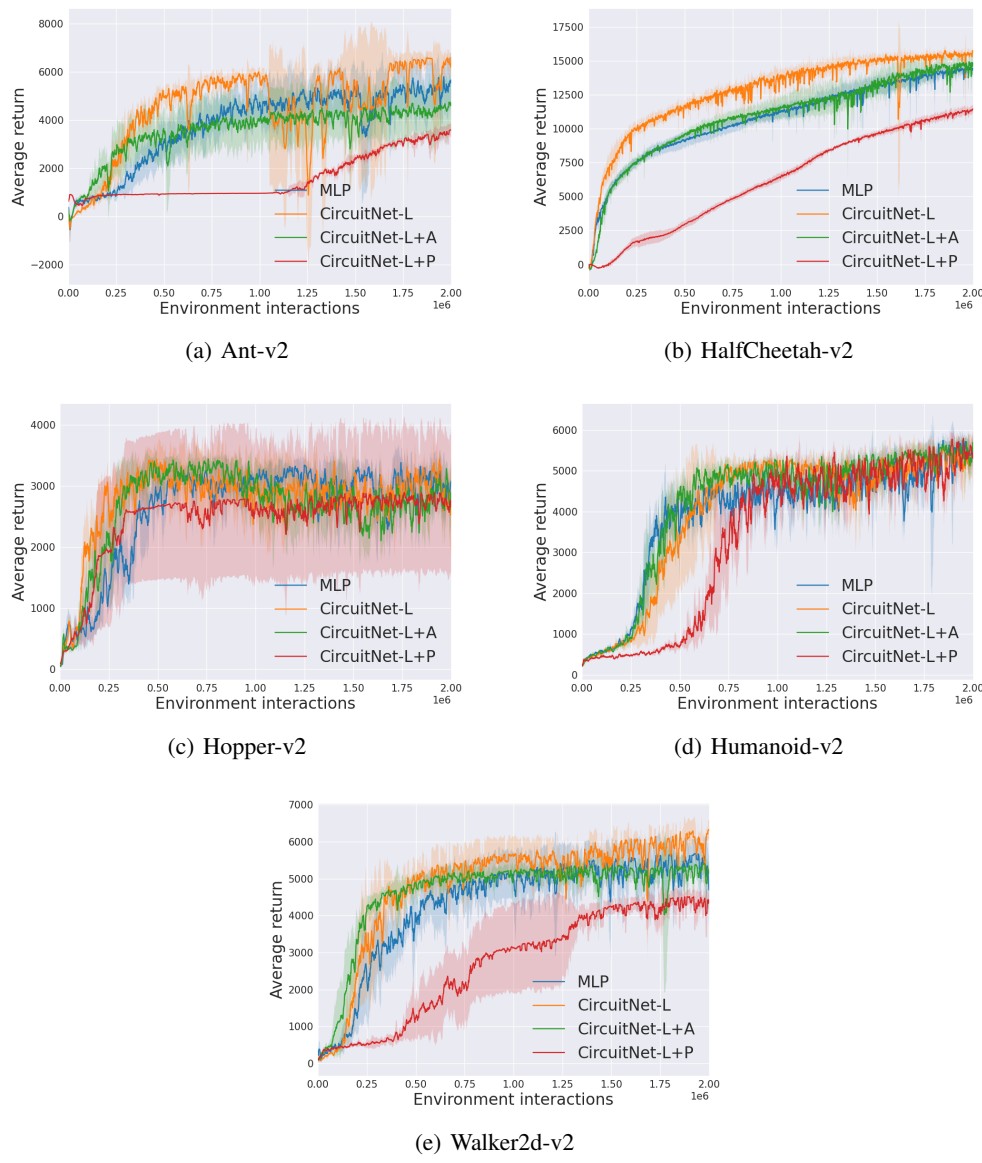

Figure 7: Learning curves of CircuitNet and MLP on all 5 continuous control benchmarks.

# D ADDITIONAL EXPERIMENT RESULTS AND ANALYSIS

## D.1 ADDITIONAL EXPERIMENT RESULTS OF DEEP REINFORCEMENT LEARNING IN MORE ENVIRONMENTS

We present results on all 5 environments in Figure 7. For the additional two environments, our model outperforms the baseline on Walker2d-v2 and achieved comparable results on Humanoid-v2.

## D.2 ABLATION STUDIES

In order to investigate the effect of modeling different types of motifs in real tasks, we compared 3 variants of CircuitNet in our experiments, (R-)CicuitNet-L, (R-)CicuitNet-L+A, and (R-)CicuitNet-L+P. The transition functions used for intra-unit signal transmissions and motifs to model are listed in Table 11.

Table 11: Description of different CircuitNet variants. $N$ is the number of neurons per Unit

| Variant of CircuitNet | Intra-unit Signal Transmisson | Motifs to Model | # Parameters per CMU |
|---|---|---|---|
| (R-)CircuitNet-L | Linear | Feed-forward, Mutual | $N^2$ |
| (R-)CircuitNet-L+A | Linear, Neuron-wise attention | Feed-forward, Mutual, Feed-back | $4N^2$ |
| (R-)CircuitNet-L | Linear, Product between Neurons | Feed-forward, Mutual, Feed-back, Lateral | $N^3 + N^2$ |

Table 12: Part of the experiment results on CIFAR-10 and CIFAR-100.

| Model | Accuracy | |
|---|---|---|
| | CIFAR-10 | CIFAR-100 |
| Best of compared baselines | 0.870 | 0.570 |
| CircuitNet-L | 0.858 | 0.588 |
| CircuitNet-L+A | 0.853 | 0.588 |
| CircuitNet-L+P | 0.913 | 0.649 |

Table 13: Part of the experiment results on time series datasets with different forecasting horizons (6, 24).

| Model | Solar-energy | | | | Pems-bay | | | | Metr-la | | | |
|---|---|---|---|---|---|---|---|---|---|---|---|---|
| | 6 | | 24 | | 6 | | 24 | | 6 | | 24 | |
| | RRSE↓ | R2↑ | RRSE↓ | R2↑ | RRSE↓ | R2↑ | RRSE↓ | R2↑ | RRSE↓ | R2↑ | RRSE↓ | R2↑ |
| Best of compared baselines | 0.202 | 0.960 | 0.355 | 0.889 | 0.452 | 0.782 | 0.532 | 0.733 | 0.517 | 0.762 | 0.692 | 0.548 |
| R-CircuitNet-L | 0.216 | 0.956 | 0.382 | 0.861 | 0.378 | 0.869 | 0.545 | 0.730 | 0.450 | 0.818 | 0.692 | 0.570 |
| R-CircuitNet-L+A | 0.212 | 0.957 | 0.409 | 0.841 | 0.377 | 0.870 | 0.561 | 0.712 | 0.448 | 0.820 | 0.667 | 0.600 |
| R-CircuitNet-L+P | 0.211 | 0.958 | 0.515 | 0.748 | 0.380 | 0.868 | 0.563 | 0.711 | 0.447 | 0.820 | 0.656 | 0.613 |

We attached some results from Table 3 and Table 5 here for convenience.

- Deep Reinforcement learning (Figure 7): both Circuit-L and Circuit-L+A achieved better or comparable results to the MLP baseline. However, with product between neurons, CircuitNet fails in DRL tasks. This may be explained by that the overly small number of neurons in CMU (in order to achieve comparable number of parameters) conversely harms the expressiveness of the network.

- Image Classification (Table 12): on Cifar-10/Cifar-100 dataset, Circuit-L and Circuit-L+A perform similarly with ResNet-18 (He et al., 2016) of comparable number of parameters and FLOPs. However, when product between neurons are added as a transition function, Circuit-L+P has a significant improvement on performance with comparable FLOPs and less parameters.

- Time Series Forecasting (Table 13): All the varients can achieve better results than the baselines on at least half of the settings. However, we don't observe a consistent trend which variant of CircuitNet can dominate the others among all the datasets.

In conclusion, even if the transition function is a linear function, the design of local density and global sparsity helps CircuitNet achieve a superior or comparable performance to the baselines. As for the comparison among variants, there isn't a consistent conclusion which is the best due to the trade-off between capturing more interactions among neurons and involving more neurons while maintaining a similar number of parameters.

## D.3 LIMITATIONS AND POSSIBLE DIRECTIONS FOR IMPROVEMENTS

Recent years, there are an enormous number of works proposed to improve the performance of neural networks in different tasks and some of them have achieved state-of-the-art performances. Quite a few branches of techniques have been proved to be very effective and widely used in the machine learning community, however, considering the fact that they are almost completely orthogonal to CircuitNet, we neither used them in CircuitNet implementations nor listed them in our experiments. These works include large-scale pre-training, advanced data augmentation, automatic neural

architecture search, etc. We optimistically expect a further improvement with exquisite task-specific designs and these advanced techniques.

