# OpenReview forum: "CircuitNet: A Generic Neural Network to Realize Universal Circuit Motif Modeling"
_ICLR.cc/2023/Conference — Submitted to ICLR 2023_

### Official Review · Reviewer_zeXY · 2022-10-25

**Confidence:** 4
**Correctness:** 3
**Technical Novelty And Significance:** 2
**Empirical Novelty And Significance:** 2
**Recommendation:** 6

**Clarity, Quality, Novelty And Reproducibility:**

The text is well-written including a comprehensive methods section. The experiments are well-designed including the usage of standard datasets/tasks and milestone baselines; however, the more recent models with higher scores need to be at least mentioned.

**Strength And Weaknesses:**

The goal of this paper is important, as there is growing evidence that the sparsity/modularity of neural network models may lead to improved performance with less computation. The authors have performed a rigorous evaluation of their model’s performance on various datasets where they compared it to the performances of multiple baseline models, published previously.

Among the others, the strengths of this submission are:

-The paper contains a comprehensive Methods appendix which includes thorough descriptions of datasets and models. The baseline models are thoroughly referenced.

-In the synthetic dataset for the function approximation, the Authors use a large enough class of the functions to be approximated, which goes beyond simple linear or quadratic functions.

-CircuitNet is compared to milestone models in vision (e.g. ResNet, ViT) and time series forecasting (e.g. ARIMA, etc.) tasks. Those models include other sparse, modular approaches (e.g. ViT) that naturally have a similarly small number of parameters.

While the strength of the proposed approach is in using a single architecture to outperform multiple baselines, meticulously optimized by previous Authors, it should be noted in the text that the current SOTA is higher than the reported performance on the used datasets. For example, on ImageNet it’s 91% and not the ballpark of 78% (https://paperswithcode.com/sota/image-classification-on-imagenet); on CIFAR-10 it’s above 99% (https://paperswithcode.com/sota/image-classification-on-cifar-10) etc. Understandably, these results required bells and whistles specific to each of the said datasets and do not devalue the results in this paper – though they still need to be mentioned and discussed not to mislead the paper readers.


**Summary Of The Paper:**

In this paper, the authors propose a locally dense, globally sparse alternative to the feedforward neural network architecture. The proposed architecture consists of interconnected blocks where the neurons within each block are connected via feedforward, attention, and product-between-neurons mechanisms. The authors compare the performance of their model to that of the feedforward networks in function approximation, reinforcement learning, computer vision, and time series forecasting tasks, where they report an increased performance compared to the baseline models of their choice.

**Summary Of The Review:**

The paper addresses the important task of improving the deep learning models’ performance and resource usage by introducing sparse and modular architecture. The Authors have compared their model to a variety of classic baseline approaches on multiple standard datasets, showing competitive performance. At the same time, the model’s scores fall below current leading scores for the used datasets (though, those likely required finetuning specific to each dataset) – this needs to be mentioned in the text. For these reasons, I recommend borderline acceptance.

---

> ### Author Response · Authors · 2022-11-14
> **Response to Reviewer zeXY**
>
> ### Comparison to state-of-the-art models
> > Q: The more recent models with higher scores need to be at least mentioned.
>
> Thanks for pointing this out! In our paper, we were not trying to claim the state-of-the-art performances on all of the tasks we listed. We can't agree more that a state-of-the-art result requires exquisite designs and fine-tuning on the datasets, and it's noteworthy that there are many techniques left to be tested in CircuitNet which have been widely applied in the machine learning community including large-scale pretraining, advanced data augmentation, automatic neural architecture search, etc., not to mention the bags of task-specific tricks. These techniques are completely orthogonal to our work and we optimistically expect a further improvement with them.
>
> We added a section (Appendix D.3) to discuss the limitations of CircuitNet and possible directions to improve the results.

---

> > ### Comment · Reviewer_zeXY · 2022-11-16
> > **Re: response**
> >
> > Thank you for your response and diligence in addressing the reviewers' comments.
> > In Tables 2-5, please add the SOTA results for the datasets used so that the paper readers won't have a wrongful idea about what the SOTA is. You may then add an argument that you aimed to compare your models with milestone baselines with a similar number of parameters / FLOPs, which is totally fine.

---

> > > ### Author Response · Authors · 2022-11-18
> > > **Re: Re: response**
> > >
> > > Thank you for the suggestion! We added lines to Table 2-5 to demonstrate the state-of-the-art results and an argument in Section 4 for that. Please kindly refer to the updated revision.

---

> > > > ### Comment · Reviewer_zeXY · 2022-11-18
> > > > **Re**
> > > >
> > > > Great, thanks for addressing the comment. At this point, I do not have any further comments/suggestions.

---

### Official Review · Reviewer_CjTa · 2022-10-25

**Confidence:** 3
**Correctness:** 3
**Technical Novelty And Significance:** 3
**Empirical Novelty And Significance:** 3
**Recommendation:** 6

**Clarity, Quality, Novelty And Reproducibility:**

Clarity:
The paper is well written and organised.  Experiment section may benefit from more implicit label explanation in +a and +p, and a more in depth centralised result discussion.

Quality:
The paper technically sounds correct and claims well supported by theoretical analysis and experimental results. Multiple experiments are conducted to support the claims.

Novelty:
To my knowledge, the idea proposed is novel.


**Strength And Weaknesses:**

Strength:
* The paper technically sounds correct and claims well supported by theoretical analysis and experimental results.
* Related works and background knowledge related are covered and discussed.
* Experiments are conducted extensively in comparing the proposed CMU net against other networks in a variety of tasks.

Weekness:
* The clarity can be further improved. For instance, the experiment section can add a table or explain in the comments on the difference between CMU, CMUL+A and +P.
* Experiment section can benefit from a small summary of why CMU can outperform in many different tasks while maintain very small size.

Questions:
* While the learning curve compares CMU against MLP in training speed, how much more or less training computing cost per iteration does CMU need compare to traditional ANNs?
* For eqn 3 (attention), what doe value, key and query represents?
* For the clipsin activation function, the appendix suggests it is somewhat inspired by leakyRELU. Is it possible to have the [-3,3] linearly for less computation cost?
* Can the authors clarify how does CMU handle conv layers, I did not quite follow the part in 3.2.

**Summary Of The Paper:**

This paper proposes CMU, a motif unit formed by a group of densely connected neocons, which is inspired by the structure of human brain. Experiment results suggests better performance.

**Summary Of The Review:**

Overall, I really like the idea of CMU and its potential of providing a more robust NN unit to build more capable networks. The experiment conducted in varies types of applications backs the idea that CMU net can be more accurate and maintain a reasonable size given it follows the design rule of locally dense and globally sparse. The paper feels most robust and novel. Clarity in explaining some CMU operation and experimental discussion can be slightly improved.

---

> ### Author Response · Authors · 2022-11-14
> **Response to Reviewer CjTa**
>
> ### Clarity of the experiment part
> > Q: The clarity can be further improved. For instance, the experiment section can add a table or explain in the comments on the difference between CMU, CMUL+A and +P.
> >
> > Q: Experiment section can benefit from a small summary of why CMU can outperform in many different tasks while maintain very small size.
>
> Thanks for your suggestion! we added a section which summaries the results of our variants and includes centralized discussions in Appendix D.2 for better clarity.
>
> We restate some of the descriptions here for your information. In our paper, (R-)CicuitNet-L, L+A, L+P in Section 4 denote for the (Recurrent) CircuitNet equipped with linear transformation only, both linear transformation and neuron-wise attention, and both linear transformation and product between neurons, respectively. These 3 CircuitNet variants correspond to the intra-unit neuron signal transmissions discussed in Section 2.1. We summarize the properties of variants in the following table.
>
> Variant of CircuitNet | Intra-unit Signal Transmisson | Motifs to Model
> ----------------------|--------------------|----------------
> (R-)CircuitNet-L | Linear | Feed-forward, Mutual
> (R-)CircuitNet-L+A | Linear, Neuron-wise Attention | Feed-forward, Mutual, Feed-back
> (R-)CircuitNet-L+P | Linear, Product between Neurons | Feed-forward, Mutual, Feed-back, Lateral
>
> ### Computational cost per iteration
> > Q: While the learning curve compares CMU against MLP in training speed, how much more or less training computing cost per iteration does CMU need compare to traditional ANNs?
>
> We listed the number of FLOPs (Floating Point Operations) per iteration for the image classification tasks (Table 3, 4). As for the other experiments, all variants of CircuitNet used for direct comparisons are of comparable parameters and FLOPs with baselines. In this way, we ensure the models to have similar computational cost per iteration.
>
> ### Key, query, and value triplets in neuron-wise attention
> > Q: For eqn 3 (attention), what doe value, key and query represents?
>
> The concept of key, query, value in neuron-wise attention are directly borrowed from Vaswani et al.(2017), where the output of the attention is calculated as a weighted sum of values and the weight assigned to each value is computed by the query and the corresponding key.
>
> As for the neuron-wise attention, attentions are applied to neurons (scalars). We map the neuron states to the key, query, value triplets with three  learnable linear projections. To make it more intuitive, we change the notations of the projection matrices to $K_i$, $Q_i$, $V_i$ for key, query, and value, respectively. (See Eq. 2 and Eq. 3)
>
> ### ClipSin
> > Q: For the clipsin activation function, the appendix suggests it is somewhat inspired by leakyRELU. Is it possible to have the [-3,3] linearly for less computation cost?
>
> To be short, the core design of the ClipSin is the sinusoidal part within [-3, 3] to address the gradient vanishing or exploding problem when quadratic terms are added to the singal transmission process. This part cannot be changed to a linear function. The trick we borrow from LeakyReLU is the small slope outside the range to prevent zero gradients.
>
> A mathematical justification about why the sinusoidal design is helpful and a linear function is not can be found in Appendix C. The key idea is:
> 1. Using an activation function with a linear part (ReLU) won't be a problem when the transmission function is linear. (Appendix C.1)
> 2. When the quadratic terms are added to the transmission function, ReLU will fail because of the linearity. This is found by analyzing the gradients (Appendix C.2) and also applies to the case when we use a linear function within [-3, 3].
> 3. Design the function that works by bounding the terms of the gradiants and choose a set of hyperparameters. (Appendix C.3, C.4)
>
> ### Convolutional layer used in the image classification task
> > Q: Can the authors clarify how does CMU handle conv layers, I did not quite follow the part in 3.2.
>
> The convolutional layer is used only as a bridge between the input images and CMUs within CircuitNets. In the image classification task, we first apply a convolutional layer to get representations for different regions, and we use the representations to initialize the CMUs. After that, we conduct several iterations of signal transmission within CircuitNet and no convolutional layers are further introduced. We refined descriptions in the model design part (See Section 3.2 and Appendix B.3.1) to reduce possible confusion.
>
> *Reference:*
>
> Vaswani, A., Shazeer, N., Parmar, N., Uszkoreit, J., Jones, L., Gomez, A. N., ... & Polosukhin, I. (2017). Attention is all you need. *Advances in neural information processing systems, 30.*

---

### Official Review · Reviewer_oKif · 2022-10-25

**Confidence:** 4
**Correctness:** 3
**Technical Novelty And Significance:** 3
**Empirical Novelty And Significance:** 3
**Recommendation:** 6

**Clarity, Quality, Novelty And Reproducibility:**

This work is of high quality and seems novel.


**Strength And Weaknesses:**

Strengths
- Viewing circuit motifs as computational building blocks is an interesting and important research direction with relevance for both AI and neuroscience.
- The model presented is well tested on a number of benchmarks and performs well.

Weaknesses
- The biological inspirations of this paper (feedforward excitation/inhibition, lateral inhibition and mutual inhibition Luo, 2021) rely on the presence of Dale's law in their definition. However Dale's law is absent in these networks, making it unclear how this work connects to biology.
- It is unclear which circuit motifs underlie the performance of CMUs. There are no ablation studies, nor analysis of the CMUs that have been learned. For example it may be that neuron-wise attention is required for all of the results, in which case the "generic" nature of CMUs is a bit misleading.
- How neuron-wise attention maps on to biology is unclear.

**Summary Of The Paper:**

This work explores how circuit motifs observed in biology can be built into artificial neural networks and finds they improve performance vs standard ANNs.


**Summary Of The Review:**

An interesting and novel study with good empirical results.

---

> ### Author Response · Authors · 2022-11-14
> **Response to Reviewer oKif**
>
> ### Dale's law
> > Q: The biological inspirations of this paper (feedforward excitation/inhibition, lateral inhibition and mutual inhibition Luo, 2021) rely on the presence of Dale's law in their definition. However Dale's law is absent in these networks, making it unclear how this work connects to biology.
>
> Thanks for your comments from the biological perspective! Note that although we're inspired from Luo(2021), our work does not depend on it and is not a full replication of the connectome of human brain. There are two aspects in Luo(2021) to dicuss about the types of motifs: one is the connectomes of neurons (circuit motifs), and the other is the types of neurons (excitory and inhibitory neurons) based on the neurotransmitters - which is related to the Dale's Law.
>
> In our work, we focus more on the "connectomes" of artificial neurons. We designed CircuitNet to represent different connectivity patterns (circuit motifs). As for the types and functionality of individual neurons, we do not put extra hypothesis and follow the previous designs to simplify the signals as real numbers. The excitory and inhibitory nature are represented by the positive or negative weights learned during the optimization process, and we leave this direction of improvements for future explorations.
>
> ### Ablation studies and analysis on different CMU types
> > Q: It is unclear which circuit motifs underlie the performance of CMUs. There are no ablation studies, nor analysis of the CMUs that have been learned. For example it may be that neuron-wise attention is required for all of the results, in which case the "generic" nature of CMUs is a bit misleading.
>
> We did ablation studies among the tasks to investigate the influence of using different intra-unit transition functions to model types of motifs. In our paper, (R-)CicuitNet-L, L+A, L+P in Section 4 denote for the (Recurrent) CircuitNet equipped with linear transformation only, both linear transformation and neuron-wise attention, and both linear transformation and product between neurons, respectively. These 3 CircuitNet variants correspond to the intra-unit neuron signal transmissions discussed in Section 2.1. We summarize the properties of variants in the following table.
>
> Variant of CircuitNet | Intra-unit Signal Transmisson | Motifs to Model
> ----------------------|--------------------|----------------
> (R-)CircuitNet-L | Linear | Feed-forward, Mutual
> (R-)CircuitNet-L+A | Linear, Neuron-wise Attention | Feed-forward, Mutual, Feed-back
> (R-)CircuitNet-L+P | Linear, Product between Neurons | Feed-forward, Mutual, Feed-back, Lateral
>
>
> Due to space limit, the results and analysis of ablations are merged with the comparison between CircuitNet and baselines in the experiment section. We restate the results and findings here as follows:
>
> * **Deep Reinforcement learning (Figure 4, 7 and Section 4.2)**: both Circuit-L and Circuit-L+A achieved better or comparable results to the MLP baseline. However, with product between neurons, CircuitNet fails in DRL tasks. This may be explained by that the overly small number of neurons in CMU (in order to achieve comparable number of parameters) conversely harms the expressiveness of the network.
> * **Image Classification (Table 3 and Section 4.3)**: on Cifar-10/Cifar-100 dataset, Circuit-L and Circuit-L+A perform similarly with ResNet-18 (He et al., 2016) of comparable number of parameters and FLOPs. However, when product between neurons are added as a transition function, Circuit-L+P has a significant improvement on performance with comparable FLOPs and less parameters.
> * **Time Series Forecasting (Table 5)**: We didn't observe a consistent trend which variant of CircuitNet can dominate the others among datasets.
>
> In conclusion, different tasks require the model's ability to model different kinds of motifs.
>
> In the revised version of paper, we add this analysis to the Appendix D.2 for better clarity.
>
>
> ### Map neuron-wise attention on to biology
> > Q: How neuron-wise attention maps on to biology is unclear.
>
> The intuition of us using the neuron-wise attention is to model the feed-back motifs (Figure 1c), where the input signals are adjusted by the target neuron states. This is analogous to the design of attention, where the incoming signal (values from input neurons) is adjusted by a weight decided by the target neuron based on the key-query pairing results.
>
> *Reference:*
>
> Liqun Luo. Architectures of neuronal circuits. *Science, 373(6559):eabg7285, 2021*.

---

### Author Response · Authors · 2022-11-14
**Common reply to all reviewers, minor revisions of the paper.**

We are encouraged that you found our work novel, interesting, and recognized its potential of providing a more robust NN unit. We sincerely appreciate all of your valuable comments, and we revised our paper based on your suggestions.

The modifications (marked with blue fonts in the paper) include:
* Section 2.1, Equation 2 and Equation 3: change the notations of matrices to calculate key, query, and value to more intuitive letters.
* Section 3.2 and Appendix B.3.1: refine descriptions about the convolutional layer to map the input images to CircuitNet.
* Appendix D.2: add a section to present a centralized discussion of CircuitNet variants and overall results.
* Section 4, Table 2-5, Appendix D.3: add results of the state of the arts. Add a section to discuss the limitations of our experiments and possible directions for improvements.

---

### Decision · Program_Chairs · 2023-01-20

**Decision:**

Reject

**Justification For Why Not Higher Score:**

While all reviewers were individually weakly positive, mostly on the grounds that the work could inspire subsequent work on bio-inspired architectures, they were all also negative about the quality of the bio-inspiration actually realized in this work.  As such, I don't think this paper passes the bar for acceptance.  The general feeling that not enough people are doing this kind of work, isn't a reason to accept this particular paper.  My general advice for the authors, in light of this discussion/review process would be to avoid the awkward middle zone identified above by reviewer oKif.  To a biologist, this paper is insufficiently biological, and to an AI researcher, it isn't performing well enough to convince them to use it.


**Justification For Why Not Lower Score:**

N/A

**Metareview: Summary, Strengths And Weaknesses:**

This paper introduces the notion of building ANNs out of circuit motifs to support greater flexibility in bio/neuro-inspired architectures.  In addition, the authors propose using bio-inspired sparse connections between locally densely connected circuit motif units.

The reviews for this paper are all 6s, pointing to all reviewers finding the goals of the paper reasonable and the paper itself being clear enough, well executed, and technically correct.  However none of the reviewers believed the approach was likely to be directly impactful.

**Summary Of Ac-Reviewer Meeting:**

Given that the reviews were borderline, we had a video discussion (AC and all 3 reviewers) about whether to accept the paper.  In this conversation, reviewer zeXY communicated belief that this paper is in the best form it could basically be in.  It is clear and interesting to this reviewer.  However, the reviewer feels it is fundamentally borderline (leaning accept) given uncertainty as to whether there are practical benefits to the approach.  Also the bio inspiration is quite loose.  Reviewer CjTa found the bio-inspired architectures interesting and was weakly positive due to the potential for these ideas to inspire subsequent work.  They didn't find the actual implementation or results very impressive.  Reviewer oKif found the paper awkwardly situated between being useful (not very) and bio-inspired (also not very).  Mostly they found it exciting to see that a bio-inspired circuit-motif architecture could even be competitive with conventional AI approaches.